# Pricing indirect emissions accelerates low—carbon transition of US light vehicle sector

Paul Wolfram [1]✉, Stephanie Weber[1], Kenneth Gillingham [1,2] & Edgar G. Hertwich [1,3]

Large–scale electric vehicle adoption can greatly reduce emissions from vehicle tailpipes. However, analysts have cautioned that it can come with increased indirect emissions from electricity and battery production that are not commonly regulated by transport policies. We combine integrated energy modeling and life cycle assessment to compare optimal policy scenarios that price emissions at the tailpipe only, versus both tailpipe and indirect emissions. Surprisingly, scenarios that also price indirect emissions exhibit higher, rather than reduced, sales of electric vehicles, while yielding lower cumulative tailpipe and indirect emissions. Expected technological change ensures that emissions from electricity and battery production are more than offset by reduced emissions of gasoline production. Given continued decarbonization of electricity supply, results show that a large–scale adoption of electric vehicles is able to reduce $CO_2$ emissions through more channels than previously expected. Further, carbon pricing of stationary sources will also favor electric vehicles.

[1] Yale University, School of the Environment, New Haven, Connecticut, USA. [2] Yale University, School of Management, New Haven, Connecticut, USA. [3] Norwegian University of Science and Technology, Department of Energy and Process Engineering, Industrial Ecology Programme, Trondheim, Norway. ✉email: paul.wolfram@aya.yale.edu

Global transportation is the single largest energy user and energy–using emitter of $CO_2$ emissions, chiefly driven by light duty vehicles (LDVs)[1]. In order to curb emissions, many countries, including the United States (US), are increasingly promoting alternative fuel vehicles, which are typically characterized by lower tailpipe emissions. However, concerns over potentially growing emissions from energy production and vehicle manufacturing have been voiced[2–5]. These emissions occur off–site, or indirectly, and include generation of electricity to charge electric vehicles, in this work ~66–86 g $CO_2$ per electric–vehicle km driven in 2020, as well as the production of vehicles, here ~16–38 g $CO_2$ per vehicle–km driven in 2020 (Supplementary Figs. 1, 2 and Supplementary Table 1). It has only recently been recognized that the emissions for producing gasoline can range significantly, from below 15 to ~320 g $CO_2$/kWh in 2015[6,7], compared to direct emissions (synonymous with tailpipe emissions) of about 250 g $CO_2$/kWh. Taken together, indirect emissions accounted for ~26% of the 1.5 Gt $CO_2$ caused by the US LDV fleet in 2020 (Supplementary Table 2). The US EPA defines LDVs as passenger vehicles and light trucks with a gross vehicle weight of up to 8500 pounds (about 3855 kg)[8].

The introduction of the Low Carbon Fuel Standard in California, which regulates all fuel and electricity production and combustion emissions, shows that transport policy in practice can at least partly address indirect vehicle emissions. However, not a single transport policy exists to date that consistently regulates all sources of vehicle emissions along the entire supply chain. Note that we use the terms 'supply chain emissions' and 'life–cycle emissions' synonymously. Both are defined as the sum of direct, or tailpipe, emissions, and indirect emissions. Fully regulating all emissions, for example through pricing, could significantly change the relative costs of different vehicle propulsion options, such as battery electric vehicles (BEVs) versus hydrogen fuel cell electric vehicles (HFCEVs) versus internal combustion engine vehicles (ICEVs). Changing costs, in turn, could affect production decisions of vehicle manufacturers, and purchase behaviors of consumers[9]. The potential impact of these relationships is unknown to date because neither model calculations, nor real–world policies, have fully accounted for or priced indirect vehicle emissions.

Integrated energy models (IEMs)[10] show that it will be challenging to reduce emissions rapidly and far enough to reach the Paris goal[11–14]. However, there is concern that IEMs do not fully represent the impact of changes in one sector, such as electricity generation technologies, on emissions in other sectors, such as industry or fuel supply[10,15–17]. For electricity generation, this has been investigated[18–23], but not for vehicles. Although global IEMs are the main tool for identifying optimal climate change mitigation pathways, they generally do not offer the same level of technological detail as national models do[10,15,16,24] which may limit their ability to identify optimal solutions across the range of options available in the real world. Further, while some integrated assessments account for materials used in electric power plants[25,26], others point out the importance of considering efficient use of resources within integrated climate scenarios[27]. Yet, material and resource efficiency have not been thought of as pollution mitigation strategies in large–scale integrated energy scenarios and are therefore not well represented in the assessment reports of the Intergovernmental Panel on Climate Change[28].

Here, we address these knowledge gaps by applying a conceptual framework by Creutzig et al.[29], which focuses on energy–demand side (rather than energy–supply side) solutions to climate change mitigation. We apply this framework to develop a comprehensive climate change mitigation analysis of an important demand–side sector, the US LDV sector. We illustrate a set of climate–change mitigation scenarios, primarily for the vehicles sector, but also consider responses in important upstream sectors, such as changes to material production, vehicle manufacturing and electricity generation. Our model of choice is the Energy Information Agency's National Energy Modeling System (NEMS). NEMS is the federal government's main tool for evaluating energy and climate policies integrative of all energy demand and supply sectors. The responses described above are normally not fully captured in NEMS. Thus, we soft–link NEMS to a detailed vehicle life cycle assessment (LCA) model (Fig. 1 and Supplementary Fig. 3). Among IEMs, NEMS has the advantage of representing the US passenger vehicle sector and its upstream sectors in great detail[10] (also see Methods), which is a prerequisite for accurately accounting for all vehicle emissions across the

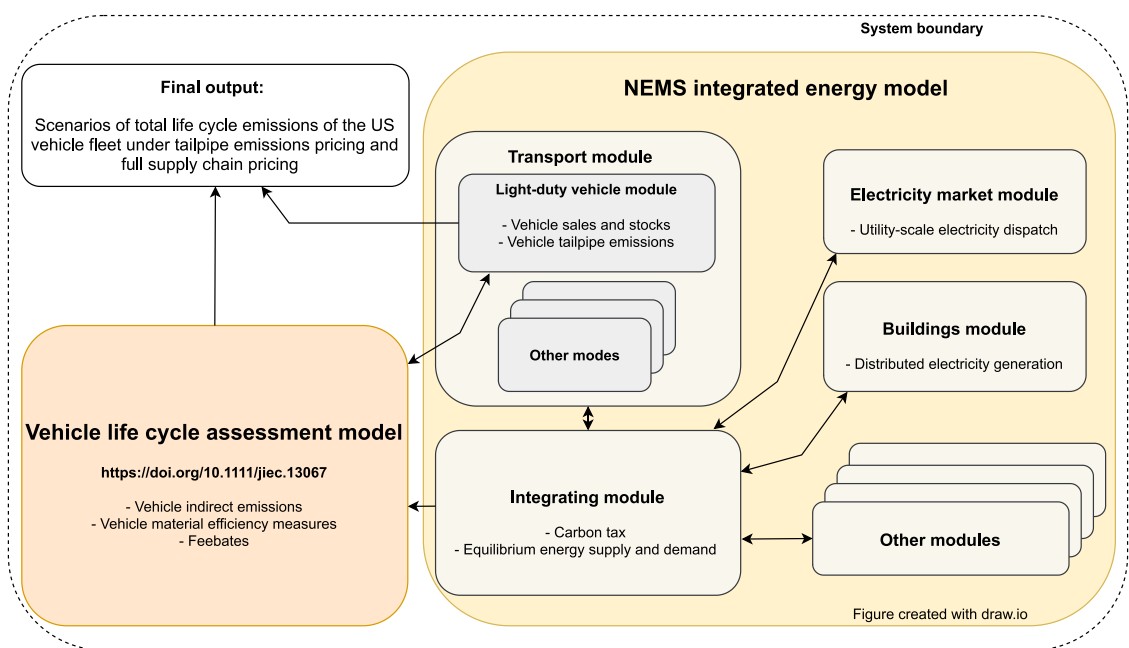

**Fig. 1 Simplified representation of linking a life cycle assessment model to the National Energy Modeling System (NEMS).** See Supplementary Fig. 3 for a more detailed model representation.

entire supply chain of the vast portfolio of available technology options. This interdisciplinary approach enables us to fully account for, and price, all life–cycle emissions that are directly (within the vehicle sector) and indirectly (in other sectors) caused by US passenger vehicles. We investigate whether this holistic emissions pricing influences the assessment of the benefit of competitive technologies. We assume that the production cost of electric vehicle batteries and renewable electricity generators fall quickly, in line with recent estimates. We further introduce a carbon price in the transport sector in 2021 which linearly increases up to 150 USD/t $CO_2$ (constant 2016$) by 2050 (Supplementary Table 3). This level is required for an LDV fleet commensurate with the US nationally determined contribution under the Paris Agreement (see Methods). For simplicity and to provide insight, we run our cases with no carbon pricing on other sectors. The difference between the two main scenarios is that either emissions from (1) the tailpipe, or (2) the entire vehicle supply chain are accounted for and priced. The implications are both surprising and significant. The strongest effect of pricing both tailpipe and indirect emissions is that the system would be pushed to an even faster phase–out of gasoline–powered vehicles, leading to a scenario minimizing both tailpipe and indirect emissions.

## Results

**Optimal vehicle choice.** While pricing only direct tailpipe emissions already leads to a nearly complete phase–out of ICEVs (Fig. 2a), the transition is accelerated under full emissions pricing (Fig. 2b). In addition, HFCEVs are avoided entirely under full pricing due to the high emissions penalty of producing hydrogen from natural gas. Lower sales of ICEVs, HFCEVs, and other powertrains (mostly hybrids and flex–fuel vehicles running both on conventional liquid fuels and biofuels) are compensated by higher BEV sales. This substitution pattern peaks around 2040 with about 2.4 million units per year (Fig. 2c). In absolute terms, sales of ICEV light trucks are reduced the most, and compensated by BEV cars and trucks. The cumulative amount of avoided

ICEVs and HFCEVs amounts to nearly 29 and 9 million units. We explore a range of side cases in which (a) only energy–chain emissions (synonymous with well–to–wheel emissions) instead of full life–cycle emissions are priced, (b) hydrogen production becomes carbon–neutral by 2050, (c) HFCEVs become cost–competitive with BEVs, as well as different combinations thereof. We display three of these cases in Supplementary Figs. 4–6. The full list of analyzed scenarios is available in Supplementary Section 5.

The mentioned substitutions of technologies lead to substantially lower cumulative life–cycle emissions through 2050 (−1.6 Gt $CO_2$, Fig. 3a, f), largely driven by lower fuel combustion (−1.4 Gt $CO_2$, Fig. 3b, g) and lower production of gasoline and hydrogen (−0.5 Gt $CO_2$, Fig. 3c, h). While stronger sales of BEVs lead to higher electricity emissions (Fig. 3d and i), these are however relatively small compared to lower emissions from fuel production (+0.25 Gt $CO_2$ vs. −0.5 Gt $CO_2$, Fig. 3k). Finally, since BEVs are material intensive[30], an additional 30 Mt $CO_2$ embodied in vehicle production can be observed. However, these could be more than compensated by ambitious recycling and reuse practices (+0.03 vs. −0.5 Gt $CO_2$, Supplementary Fig. 7). As mentioned earlier, we explore a range of side cases (Supplementary Figs. 4–6) which show some variation in their potential for emission reductions (also see dotted lines in Fig. 3a–j) but the overall trend is robust among these cases. Accordingly, additional cumulative life–cycle emission reductions can vary between −1.4 and −1.7 Gt $CO_2$ across all cases (see dotted lines in Fig. 3f) on top of emission reductions already achieved under pricing direct emissions only. Only in scenarios of constant renewable electricity costs does full emissions pricing not yield lower emissions than direct–emissions–only pricing (see Uncertainty analysis and Supplementary Figs. 8, 9). In Fig. 3k, the differences in emissions between 'full pricing' and 'direct–emissions–only pricing' are once more plotted by life cycle stage, while in Fig. 3l, all sources of indirect emissions, i.e., production of fuels, electricity, and vehicles, are categorized as such. It becomes apparent that 'full pricing' not only leads to reduced tailpipe

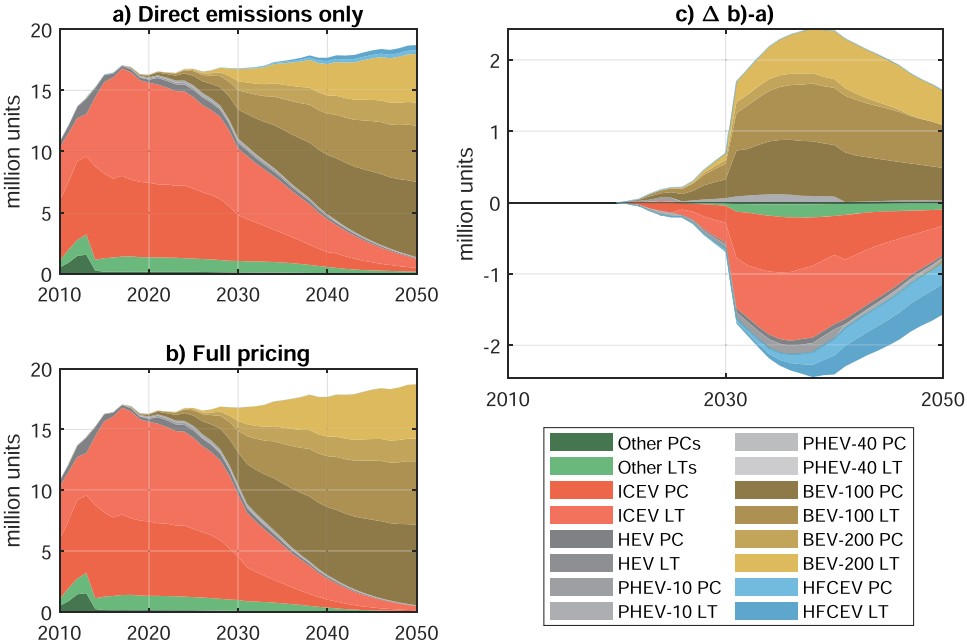

**Fig. 2 Optimal vehicle choice under different emissions pricing scenarios.** Direct–emissions–only pricing (**a**) and full emissions pricing (**b**). **c** Differences in vehicle choice between (**b**) and (**a**). PC Passenger car, LT Light truck, ICEV Internal combustion engine vehicle, HEV Hybrid electric vehicle, PHEV Plug–in hybrid electric vehicle, BEV Battery electric vehicle, HFCEV Hydrogen fuel cell electric vehicle; –10 10–mile electric range. The underlying data used to compile this figure can be found in Supplementary Table 4.

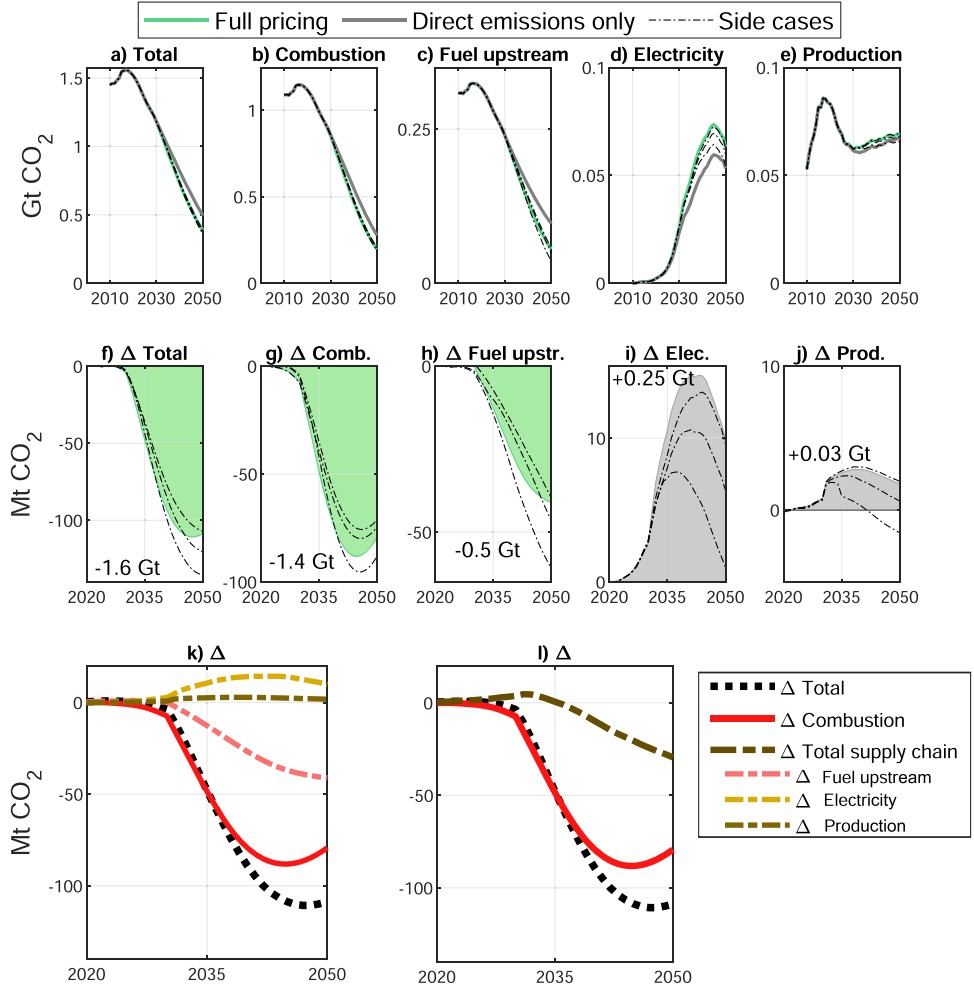

**Fig. 3 Life-cycle CO$_2$ emissions of the US light vehicle fleet when fully pricing emissions ('Full pricing') and when only pricing direct emissions ('Direct emissions').** Total emissions (**a**) and broken down by life-cycle stage (**b-e**). Differences in emissions between full and direct-emissions-only pricing (**f-l**). Dotted lines (**a-j**) illustrate results from side cases ('Side cases', also see Supplementary Figs. 4-6). The underlying data used to compile this figure can be found in Supplementary Table 2.

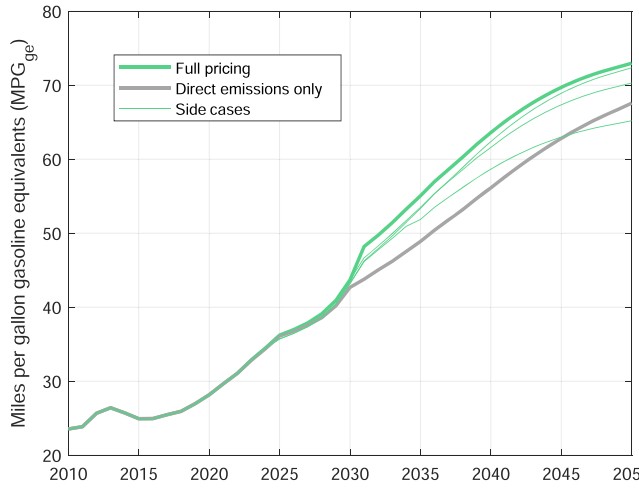

**Fig. 4 Average real-world fuel economy of the US light vehicle fleet when all emissions are priced ('Full pricing') and when only direct emissions are priced ('Direct emissions only').** The thin green lines show the range of results from the side cases ('Side cases'). The underlying data used to compile this figure can be found in Supplementary Table 5.

emissions but also to lower indirect supply chain emissions, at least after about 2035.

**Fleet efficiency.** The future of the Corporate Average Fuel Economy (CAFE) standard is currently highly uncertain and we, therefore, do not model changes to CAFE after 2025. While the Trump administration enacted the Safer Affordable Fuel-Efficient (SAFE) standard in 2018, which weakened CAFE requirements through 2026, the Biden campaign announced a plan to consider a more ambitious CAFE[31]. While further details were unknown at the time of producing the results, a new proposal has been announced on August 5, 2021, which foresees an improvement of the average fleet-wide fuel economy of new 2026 vehicles by 12 miles per gallon relative to 2021 vehicles[32]. Despite the fact that CAFE is not further tightened after 2025 in our model, average real-world fuel economy?[33] of the fleet continues to improve greatly in all scenarios, even after 2025 (Fig. 4). This can be explained by the strong market penetration of BEVs. When full life-cycle emissions are priced, average fuel economy is even higher compared to direct-emissions-only pricing due to the accelerated penetration of BEVs. Side cases with higher shares of HFCEVs however exhibit significantly lower average fuel

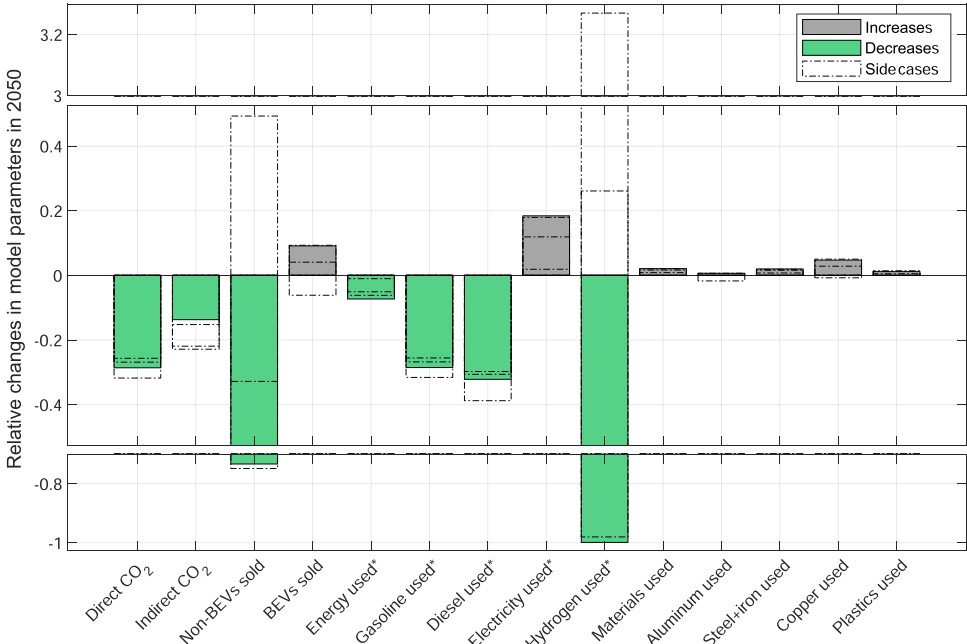

**Fig. 5 Changes in model parameters ('Increases' and 'Decreases') due to full pricing relative to direct–emissions–only pricing in 2050 (green and grey colored bars).** The hollow bars show the normalized differences of the side cases ('Side cases') relative to direct–emissions–only pricing in 2050. BEV Battery electric vehicle; * Onboard energy. The underlying data used to compile this figure can be found in Supplementary Table 7.

economies (Supplementary Table 5). Other fleet characteristics, such as average vehicle weight, deployment of lightweight through material substitution, segment shares, and total travel demand, are less impacted in the different scenarios (Supplementary Fig. 11).

**Resource use implications.** Fully pricing life cycle emissions would also have important implications on resource use (Fig. 5). For example, gasoline and diesel consumption in 2050 would be 29% and 32% lower than if taxing only direct emissions (26–32% and 30–39% in the side cases). Further, while hydrogen demand would be lower by 99.9% (–98.0 to +327.0%), electricity use would be 18% higher (2–18%). Overall, taxing supply chains would lower energy use by 7% (1–6%). Meanwhile, overall material demand would be slightly higher, by 2.1% (0.8–2.0%), with copper demand higher by 4.7% (with a range of −0.7 to +5.0%). These effects are largely due to higher BEV sales (by 9.1%, with a range of –6.1% to +9.3%, Supplementary Table 6).

In absolute terms, the difference between the two main scenarios is that under full pricing, cumulative gasoline consumption through 2050 is reduced by 0.15 trillion gallons or 0.6 trillion liters compared to direct–emissions–only pricing. This corresponds to the US gasoline consumption of a whole year (about 0.12 trillion gallons or 0.5 trillion liters in 2020)[34]. Similarly, 0.8 PWh (trillion kWh) of hydrogen is saved, while electricity consumption is higher by 3.2 PWh, roughly corresponding to the current annual amount of electricity end–use in the US in 2019[35] (Supplementary Table 8). Cumulative material use for vehicle production is moderately higher, by about 24 Mt, which is chiefly driven by the larger stock of material–intensive BEVs. The higher material demand is partially mitigated by a slightly lower average vehicle weight (Supplementary Fig. 11). Material substitutions due to vehicle lightweighting lead to marginally higher use of aluminum and plastics, and a reduction of steel and iron. Overall, the largest absolute difference is due to the higher demand for stainless steel (+13.2 Mt), followed by copper (+3.1 Mt), aluminum (+1.6 Mt), and plastics (+1.2 Mt).

Simultaneously, the use of automotive steel and cast iron is lower, by 1.0 Mt. However, more ambitious recycling and reuse practices have the potential to more than offset the stronger demand for virgin materials, by about 740 Mt (Supplementary Fig. 7).

**Adequacy of analyzed decarbonization measures.** Emission reductions required to halt climate change are sometimes framed through the carbon budget — the amount of emissions remaining until the atmosphere reaches an identified temperature threshold. Under the described cases, the US LDV sector would require 3–5% of the global carbon budget identified by the Intergovernmental Panel on Climate Change[14], which is about as much as its share of current emissions (Supplementary Table 2).

## Discussion

There is remarkably little known about the extent to which indirect emissions shape cost–optimal decarbonization pathways[36] and vice versa. Previous work focusing on electricity supply reported a limited role of indirect emissions in optimized climate change mitigation scenarios[18–20]. In this work, we explored the role of indirect emissions in the decarbonization efforts of the US passenger vehicle fleet and find that, in fact, they can significantly alter optimal climate change mitigation pathways. An important difference between the electricity supply sector and the LDV sector is that indirect emissions play a larger role, accounting for about a quarter of total life–cycle emissions already today. In our scenarios, indirect emissions make up almost half of total LDV sector $CO_2$ emissions in 2050 (44–49%) and about 24–29% of cumulative emissions over the 2010–2050 scenario time frame (Supplementary Table 2). For comparison, McDowall et al. report that indirect emissions would account for less than 10% of total life–cycle power plant emissions in 2050 in an optimal decarbonization scenario of the EU[19].

Although overall life–cycle emissions are significantly lower under full emissions pricing, the share of indirect emissions increases, most prominently due to electricity generation and battery manufacturing for BEVs. However, higher electricity

emissions are more than offset by lower gasoline supply–chain emissions stemming from the production of crude oil (Fig. 3k). Higher emissions from material production and vehicle assembly are relatively small and could be more than offset by increased material efficiency efforts including more ambitious material recycling and reuse of components.

While it is expected that direct emissions of BEVs are lower than those of ICEVs, it is surprising that in fact non–tailpipe emissions are also lower (Fig. 3l). This sheds new light on the current public debate about 'dirty' batteries and electricity[37]. In fact, the simultaneous reduction of both direct and indirect emissions indicates a win–win situation for climate change mitigation, meaning that climate policy with very high shares of BEVs represents a no–regrets strategy in terms of emissions (but only if electricity continues to decarbonize as has been assumed in our main scenarios). Our insights are therefore highly relevant for global climate and transport policies. Current policies, such as performance standards or emission pricing schemes, should be broadened in their scope in order to regulate all sources of vehicle emissions along the entire supply chain or throughout the entire life cycle. Our scenarios further indicate that the US (and likely other nations with suitable low–carbon electricity grids) should target the deployment of BEVs. HFCEVs could offer a viable alternative if costs to produce fuel cells and low–carbon hydrogen would fall considerably in coming years.

Our work represents a step towards a holistic inclusion of dynamic life–cycle relationships in integrated modeling frameworks. Future research could include additional potentially important factors and processes, such as the deployment of carbon capture and storage (CCS) at fuel refineries, differences in emission intensities of hydrocarbons, synthetic liquid fuels, net–negative emission pathways of energy production, and low–carbon steel production using hydrogen from renewable sources. Future research could also investigate the degree to which our results would differ in various regions of the world, or if additional pollutants, other than direct and indirect $CO_2$ emissions, were internalized in optimal pollution mitigation pathways.

## Methods

**Demand–side framework**. We address calls for stronger research focus on demand–side solutions for mitigating climate change[38]. Specifically, we apply and specify a transdisciplinary demand–side assessment framework focusing on an important emitting sector[29] (Supplementary Figure 3). Our framework addresses the following key areas: (1) End–use context: we focus on demand–side solutions, with the US LDV fleet as a case study. (2) Technology: we use industrial ecology methods to model full life cycle $CO_2$ emissions and costs of all major established and emerging vehicle technologies. This enables us to test the potential of different technological mitigation measures along the entire vehicle supply chain including powertrain switching, changes in material composition, recycling of materials, reuse of vehicle components, and feedstock switching for fuel and electricity production. (3) Policy instruments: Carbon pricing is applied to either tailpipe emissions or the entire vehicle life cycle. (4) Climate change mitigation pathways: We present climate change mitigation scenarios of the US LDV sector and analyze the contribution of several mitigation measures towards the US nationally determined contribution and a 2 °C consistent US LDV sector. (5) Sustainable development: We highlight synergies with other sustainability indicators such as resource use, energy use, and consumer cost.

**Integrated energy modelling**. Our tool of choice is NEMS which is the model behind the well–known Annual Energy Outlook[39,40]. In this study, we use the NEMS code run on a server at Yale University (henceforth we call it 'Yale–NEMS' at EIA's request) and slightly modified to output additional results[41,42]. Yale–NEMS sets prices so that an equilibrium is obtained where annual energy supply equals energy demand (in each energy market) through 2050. The main energy demand sectors are residential buildings, commercial buildings, transport, and industry. Projections of economic drivers are provided exogenously while world energy prices, world energy supply and demand, and US energy imports and exports are calculated endogenously. Yale–NEMS provides a full account of $CO_2$ emissions across all industries and a range of air pollutants from vehicles and power plants. $CO_2$ accounted for 97% of total GHG emissions of the US electricity

and transport sectors in 2019[43]. Other GHGs such as methane emissions from fossil–fuel extraction and hydroelectric power plants are not included.

The transport sector includes several modes of travel, such as LDVs, aviation, trucking, shipping, and rail. The LDV submodule distinguishes twelve vehicle sizes, 86 fuel efficiency technologies, as well as sixteen alternative propulsion technologies including BEV–100 (100 mile electric range), BEV–200, PHEV–10, PHEV–40, HEV, and HFCEV. Various fuel pathways are modeled as well. The LDV submodule uses a discrete choice formulation to simulate both the behavior of vehicle manufacturers and consumers. Consumers base purchase decisions on energy prices, charger and fuel station availability, vehicle purchase prices, and a range of other vehicle attributes. The decision–making process of vehicle manufacturers is usually not considered in large–scale IEMs[2]. Thus a distinguishing feature of Yale–NEMS is that vehicle manufacturers make production decisions based on technology cost, CAFE requirements, and potential regulatory costs. Further details on EIA's NEMS and a direct comparison with other IEMs can be found elsewhere[10]. Here we make several refinements to Yale–NEMS' LDV submodule: We update vehicle costs in a bottom–up fashion using detailed cost estimates for all major vehicle components, such as engines, electric motors, transmissions, fuel cells, and hydrogen storage tanks[44,45]. Further, costs of lithium–ion batteries start out at about 465 USD/kWh in 2016 and reach floor costs of ~ 83 USD/kWh over the modelled time horizon due to economies of scale and technological development (Supplementary Table 9). This cost development is within the range of recent estimates[46].

The electricity market module considers all major fossil and renewable generators, including conventional, and advanced coal and gas power plants with and without CCS, nuclear, hydro, solar thermal, solar photovoltaics (PV), and on– and offshore wind power. A dispatch model determines electricity supply, demand, and prices at the sub–annual level (three seasons by three times of day). In our scenarios, we assume that overnight capital costs of solar PV and onshore wind power plants fall from around 1245 and 1230 USD/kW in 2019 down to about 370 and 540 USD/kW by 2050 due to economies of scale and technological developments. This cost development is within the range of recent estimates[47,48]. As a result, new power plant capacities are mainly provided by renewable electricity generators, while fossil–fueled power plants retire (Supplementary Figs. 12, 14). Thus, renewables provide more than half of all electricity well before 2030 and more than three quarters by 2050, confirming a widely reproduced modelling result[26,49,50]. The remaining electricity demand in 2050 is mainly met by natural gas (16%) and nuclear power (6%), while coal is almost entirely phased out (1.5%, Supplementary Table 10). A small percentage of electricity from coal is generated at CCS–equipped plants. Electricity is not only produced in the power supply sector but also in the residential and commercial end–use sectors — a feature that sets apart Yale–NEMS from other IEMs[24] — with the main technologies being rooftop solar PV and distributed natural gas (for modelling details, see for example section 'Distributed Generation and Combined Heat and Power (CHP) Submodule' within the 'Commercial Demand' section of the NEMS documentation[40]). While electricity demand grows from almost four to more than six trillion kWh, an increase by more than half, electricity emissions fall from almost 2400 to below 290 Mt $CO_2$, a reduction of 88%. As a result, the carbon intensity of the electricity mix falls by a factor of twelve, from 546 down to 45 g $CO_2$/kWh (Supplementary Fig. 13 and Supplementary Table 10). Since we focus on the accounting of $CO_2$ emissions, we do not take into account emissions from methane leakage during fossil fuel production.

In all scenarios, we introduce a price on carbon in the transport sector in 2021 which linearly increases up to 150 USD/t $CO_2$ by 2050 (constant 2016$, Supplementary Table 3) — a level required to meet the US nationally determined contribution under the Paris Agreement. The US is committed to reducing $CO_2$ emissions by 80% by 2050 relative to 2005. We assume that all sectors equally attempt to reduce their emissions by that percentage. According to the US EPA, direct $CO_2$ emissions from the US LDV fleet amounted to 1180 Mt in 2005[43]. The growing carbon price in the scenarios leads to a significant cost increase of energy carriers, especially gasoline (Supplementary Fig. 13). Combined with the cost reductions of electric vehicle batteries and renewable power plants, our assumed carbon price leads to reductions of direct $CO_2$ emissions on the order of 76–84% in 2050 relative to 2005, depending on the specific scenario (Supplementary Table 3).

**Soft–linking Yale–NEMS with LCA**. We soft–link Yale–NEMS to a detailed passenger vehicle LCA model[51] and iterate between the LCA model and Yale–NEMS until inputs and outputs converge between both models. The LCA model covers $CO_2$ emissions of all major technologies across the entire vehicle life cycle, including fuel production and combustion, electricity generation, material production and recycling, assembly and reuse of vehicle components, and lightweighting through material substitution. For simplicity purposes, the LCA model assumes that vehicle production takes place in the US. Furthermore, the model includes the most climate–relevant vehicle materials and disregards other minor materials (see Supplementary Section 11 for a discussion of the error invoked from these assumptions).

In a first iteration we calibrate the LCA model to the US case by using the following Yale–NEMS outputs as calibration coefficients: (1) Vehicle baseline weights (without lightweighting, Supplementary Table 11), (2) the expected degree of vehicle lightweighting (substitution of conventional materials with lightweight materials, Supplementary Table 12), (3) current and future on–road energy

consumption (Supplementary Table 11), (4) current and future battery sizes (Supplementary Table 11), (5) current and future carbon intensity of electricity generation used to manufacture vehicles and charge BEVs (Supplementary Table 10), and (6) carbon prices (Supplementary Table 3). Taking these variables into account, the LCA model calculates per–vehicle life–cycle carbon emissions (Supplementary Section 2) and translates these into life–cycle carbon prices for each technology (Supplementary Section 4 and Supplementary Table 13).

The obtained carbon prices are then linked back to Yale–NEMS for consideration in the vehicle choice procedure of the LDV submodule. Specifically, carbon prices on indirect emissions are implemented in Yale–NEMS as a so–called 'feebate'. Feebates are regarded as an effective policy instrument to reduce vehicle emissions in the new vehicle fleet[52–54]. Feebate systems impose a fee on vehicles with high $CO_2$ emissions and grant a rebate to low–carbon vehicles. Here we apply that design to both the production of vehicles and energy carriers separately, with two main steps. First, if the production of any alternative vehicle technology $a$ is more carbon–intensive than the production of an ICEV, a fee is added to the purchase price of $a$, otherwise a rebate is granted. Second, if the production of the energy source that is used in $a$ over $a$'s lifetime is expected to create more $CO_2$ than the production of gasoline used in an ICEV, then an additional fee is added to $a$'s purchase price, while a rebate is provided otherwise (Equations (1)–(3)). For example, a fee is imposed on the production of BEVs, largely due to the energy– and material–intensive battery. This fee is increasing with the growing carbon price (although partially mitigated by the falling carbon intensity of production), from about 9–15 USD/BEV in 2021 to about 120–210 USD/BEV in 2050, depending on vehicle and battery size. A rebate is however granted due to the production of electricity that the BEV is expected to charge over its lifetime. This rebate is growing stronger each year as electricity quickly decarbonizes — from about 400 USD/BEV in 2021 to ~2600 USD/BEV in 2050 (Supplementary Figs. 4, 5, and Supplementary Table 13). This way, these fees or credits become part of the decision–making process of vehicle manufacturers and consumers, and therefore influence both vehicle production and sales in Yale–NEMS.

$$F_{a,y}^{V} = (E_{a,y}^{V} - E_{g,y}^{V}) \times T_y \quad (1)$$

$$F_{a,y}^{C} = \sum_{l}^{L}(E_{a,y}^{C} - E_{g,y}^{C}) \times T_y \quad (2)$$

$$P_{a,y} = P_{a,y} + F_{a,y}^{V} + F_{a,y}^{C} \quad (3)$$

where:

$F$ denotes feebate
$a$ denotes alternative vehicle
$y$ denotes year
$V$ denotes vehicle production
$E$ denotes $CO_2$ emissions
$g$ denotes gasoline–powered ICEV
$T$ denotes carbon tax
$C$ denotes vehicle energy chain
$l$ denotes vehicle age
$L$ denotes vehicle lifetime
$P$ denotes vehicle purchase price

Note that due to the large uncertainties involved we do not attempt to estimate the costs of electric vehicle chargers[45] (especially when allocating a certain fraction of the cost of public chargers to individual BEVs and PHEVs), nor do we attempt to estimate how strongly consumers would discount future costs[54]. We do however acknowledge that these factors could impact consumer choice. Since we wish to present our results in isolation of these factors we leave it for future research to quantify the influence of these effects.

In a second iteration of the LCA model, total vehicle sales by technology and segment, and total energy use by energy carrier are extracted from Yale–NEMS and fed back into the LCA model. In addition, any vehicle characteristics, such as vehicle weights and lightweighting shares, that have been altered by the life–cycle carbon price implemented in Yale–NEMS (Supplementary Fig. 3), are updated in the LCA model accordingly. As a result of the second LCA model run, total indirect emissions of vehicle and energy production over time are obtained (Supplementary Table 2). Tailpipe emissions and emissions from electricity use are taken directly from Yale–NEMS (Supplementary Table 2).

**Uncertainty analysis.** In addition to the mentioned side cases, we present a set of six scenarios to explore the uncertainty of future costs of electric vehicle batteries and renewable power generators (as well as two additional scenarios exploring uncertainties around battery density, that we discuss in Supplementary Fig. 10). Specifically, we investigate the effects of taxing direct as well as full supply chain emissions in scenarios of high costs of renewable power plants and/or EV batteries. We assume that the cost of EV batteries remain constant after 2021. This constant price can be interpreted either as insufficient investments into battery technology or as growing raw material prices. The International Energy Agency notes that "a doubling of lithium or nickel prices would induce a 6% increase in battery costs. If both lithium and nickel prices were to double at the same time, this would offset all the anticipated unit cost reductions associated with a doubling of battery

production capacity"[55]. Similarly, the overnight capital costs of wind and solar PV power plants remain at 2021 levels. The share of renewable electricity generation grows from 23% in 2021 to 31% by 2050, a mild increase, which is largely enabled by energy storage installations. If supply chain emissions are fully priced, the higher carbon intensity of electricity implies a substantial two–fold fee on BEVs: (1) A carbon fee on the production of BEVs starts at 10–16 USD in 2021 and grows up to 220–380 USD per vehicle by 2050. (2) Similarly, a carbon fee on charging BEVs increases from 380 to 400 USD in 2021 to 1070–1140 USD per vehicle by 2050. In this case, BEVs are not cost–competitive with ICEVs and therefore do not gain significant market share. If only direct emissions are priced, the transition towards BEVs is only a little perturbed. Assuming constant EV battery prices alone does not strongly affect EV sales, but leads to almost complete substitution of long–range BEVs with short–range BEVs.

In terms of emissions, a carbon tax on supply chain emissions is not able to yield the desired results if the electricity grid does not face substantial decarbonization. In this case, pricing supply chain emissions lead to higher emissions compared to pricing direct emissions only (see Supplementary Fig. 9 for more details and results).

**Reporting Summary**. Further information on research design is available in the Nature Research Reporting Summary linked to this article.

## Data availability
Relevant input data and model outputs generated in this study are provided in the Supplementary Information/Source Data file.

## Code availability
A version of the LCA model code used in this work is available in an open repository at Zenodo: https://doi.org/10.5281/zenodo.3896664.

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

## Acknowledgements

We thank Pei Huang at the ZEW – Leibniz Centre for European Economic Research, Brian Andrew Reed at Stanford University, Paul Kondis at the US Energy Information Administration for assistance with NEMS. We also thank three anonymous reviewers for their feedback. This publication was developed under Assistance Agreement No. RD835871 was awarded by the U.S. Environmental Protection Agency (EPA) to Yale University. It has not been formally reviewed by EPA. The views expressed in this document are solely those of the authors and do not necessarily reflect those of the Agency. EPA does not endorse any products or commercial services mentioned in this publication.

## Author contributions

PW designed the approach, collected the data, performed the experiments, and wrote the paper. SW assisted in setting up the experiments. KG and EH supervised the work. SW, KG, and EH helped edit the paper.

## Competing interests

The authors declare no competing interests.
