## [Peer Review File · Nature Communications]

Reviewer comments, first round review -

Reviewer #1 (Remarks to the Author):

The overall premise of the paper is interesting, though the level of novelty is perhaps not the highest.

The overall results could be more clearly described - particularly the relationship between pricing, materials and emissions towards changing the vehicle mix.

The NEMS model is, as far as I understand, a CGE model for energy consuming sectors of the US economy, but it is not clear if supply-demand effects on price that are considered for the oil market (for example) are similarly considered for producers of non-energy materials (such as lithium and lithium batteries). In the present study, how is the effect of price considered on the material sectors - particularly when many of the "other" materials are not domestically produced in the US?

There is a particular practical issue of whether offshore emissions could be effectively included in the price or not.

Likewise, in the LCA data, the "other" materials would be expected to be important - otherwise the only(?) material difference between the vehicle types with functional relevance is copper? The remainder of the materials would appear to be structural? In this case, should the vehicle chassis be unified and a clearer delineation of the impacts that arise solely due to the vehicle power train given?

An inventory of the "other" materials would be useful.

Is there any consideration of the need for additional electricity generation to power a high BEV load in transport? (i.e. the electricity demand in the sector would increase, requiring additional capacity installation - not just a change from high CO2 to low CO2 mix)

It would be useful to include a diagram showing the flows of information and steps in the model to enhance understanding of the modelling approach.

Line 36 - is this 26% additional to the 1.5Gt emissions, or included?

Line 48 - aren't these models referred to as Integrated Assessment Models, not specifically Integrated Energy Models?

Reviewer #2 (Remarks to the Author):

This paper uses a quantitative model to simulate the transition from internal combustion engine vehicles to electric vehicles. The paper uses a life cycle analysis model of vehicle inputs and an energy model to look at how carbon pricing affects vehicle adoption. The paper shows that if renewables could provide more than 75% of electricity, then a carbon price on the extraction/processing of oil and the combustion of gasoline for conventional vehicles (and a similar carbon price for all other emissions like vehicle construction and the remaining polluting electricity sources) would lead to greater EV adoption than a carbon price just on gasoline combustion. The authors conclude that: "the simultaneous reduction of both direct and indirect emissions indicates a win-win situation for climate change mitigation, meaning that climate policy with very high shares of BEVs represents a no-regrets strategy." And furthermore, that the US "should target deployment of BEVs and largely disregard competing technologies."

Main comments

1. Low prices of renewables are not sufficient for large scale adoption. Electricity reliability requires that supply equal demand at all moments. When the sun is not shining and the wind is not blowing, how does a grid with 75% renewables operate? Without explicit modelling of the very large costs of storage, additional transmission, and investments to address issues of reliability, the model likely understates the costs of, or even overstates the feasibility of, large-scale renewable

deployment. A model that looks at the grid on an annual basis will not capture these issues.

2. Are fugitive methane emissions from extracting natural gas in the model?
3. The authors' conclusions ignore uncertainty. I suggest toning down the language like "no-regrets" and "disregard competing technologies".
4. In fact, showing how sensitive the paper's results are to the assumptions on renewables penetration would be helpful. If renewables do not get cheaper and coal plants do not retire, what does the model find? I would think that the indirect emissions of extracting coal and natural gas for producing electricity (especially if fugitive emissions are high) could be similar to the range of the indirect emissions from extracting and processing crude oil.

Minor comments

5. Page 2, lines 34-35: gasoline production emissions presumably are in g CO₂/km, not kWh.
6. Page 2, line 59 EIA stands for Energy Information Administration, not Agency.
7. Nature Communications does not use footnotes.

Reviewer #3 (Remarks to the Author):

The submitted manuscript has investigated the life cycle environmental and economic implications of pricing, through the carbon tax, GHG emissions from passenger vehicles throughout their life cycles as an enviro-economical policy measure. In this pursuit, the researchers claim that pricing the GHG emissions associated with a vehicle's life cycle (though they have not provided a figure showing the system boundaries drawn for the life cycle assessment model) could be a more effective policy in reducing the environmental impacts of U.S. transportation than pricing only the GHG emissions from a vehicle's tailpipe. They further claim that such a policy measure could accelerate the phase-out of conventional vehicles while leading to higher penetration of battery electric vehicles and hence increased GHG savings.

Environmental policies have not been as efficient and effective as they are supposed to be for mitigating the negative impacts of anthropogenic activities (e.g. transportation). Pricing vehicle's life cycle emissions through carbon tax is one of such policies that is likely to have far-reaching implications in terms of the sustainability profile of U.S. passenger vehicles. Therefore, I think that gaining insights into these implications is of interest to the field of industrial ecology and others in the community and the wider field, especially given the significance of effective enviro-economical policies for mitigating the climate crisis studied in this field.

Two models - i) Yale-NEMS and ii) LCA model previously developed and published- were combined to carry out the quantitative analysis. Even though the supplementary information, including the input data, have been provided and includes data on many variables used in the models mentioned above, previously published works have been referred for some information, e.g. cost estimates for engines, electric motors, transmissions, fuel cells, and hydrogen storage tanks (and their specifications). The researchers are recommended to provide such information in a table, at least, in the SI. Furthermore, even though the source code for the LCA model has been provided, the mathematical notations of the model formulations employed in the study, as well as of the incorporation of the assumptions into these models, are missing. I would recommend that the researchers consider addressing these in the SI, at least if the word limit does not allow them to be addressed in the main manuscript. This would make it easier for the reader to have a better understanding of the analytical work done and for other researchers to be able to reproduce the results and build upon the models. Also, the researchers are recommended to provide a figure, depicting the system boundary for the LCA model developed.

I have submitted my comments as annotations and attached the annotated manuscript.

August 20, 2021

Response to the reviewers

Dear anonymous reviewers,

We would like to thank you very much for your thoughtful reviews of our previously submitted manuscript “Pricing indirect emissions accelerates low-carbon transition of US light vehicle sector”. We have worked hard to successfully address all the concerns that have been expressed and we believe that the paper has improved as a result. We look forward to hearing from you.

Kindest regards,

Paul Wolfram (on behalf of all authors)

Reviewer 1

The overall premise of the paper is interesting, though the level of novelty is perhaps not the highest.

Thank you for this thought. We appreciate that you find the overall premise interesting. In the comment on the novelty, we believe that the reviewer may be referring to the fact that previous static life cycle assessments have highlighted the importance of the electricity mix for both charging vehicles and producing the vehicles in the first place. We agree with this assessment of the literature. However, the present work differs in being the first large-scale dynamic assessment of electric vehicle roll-out scenarios considering both the life cycle coefficients *and* the dynamic effects of the entire energy system at the same time. This combination of approaches is novel and leads to a new result to the literature: that the negative impacts that occur with electric vehicle adoption can be largely avoided. Thus, we see our manuscript to be of great interest to those engaged in the lively debate about EV policy. We would like to point out that the article pre-print on ResearchSquare attracted more than 300 viewers, with 100 alone within the first five days. This perhaps can be seen as at least one indication that the broader academic community will find the work novel.

The overall results could be more clearly described - particularly the relationship between pricing, materials and emissions towards changing the vehicle mix.

Thank you for this suggestion - it is one that we have taken seriously. One new aspect of this work above the previous literature is that the pricing of all emissions, including those stemming from material production, influence the optimal vehicle fleet. Thus, we feel it is especially important to document this part of our approach clearly and we appreciate your nudge to do so. We added three mathematical equations in the methods section providing additional detail on how embodied emissions from material production and energy chains affect vehicle prices. We further added two sections in the supplementary information which serve as a more detailed description of the interconnection between prices, materials and emissions (Supplementary Sections 2 and 10). We believe that this additional information complements the existing descriptions in the main manuscript well and we thank you again for this comment.

The NEMS model is, as far as I understand, a CGE model for energy consuming sectors of the US economy, but it is not clear if supply-demand effects on price that are considered for the oil market (for example) are similarly considered for producers of non-energy materials (such as lithium and lithium batteries). In the present study, how is the effect of price considered on the material sectors - particularly when many of the ‘other’ materials are not domestically produced in the US? There is a particular practical issue of whether offshore emissions could be effectively included in the price or not.

The reviewer’s understanding of the model is correct. NEMS is indeed a CGE model focusing on the US economy. (In addition, the model includes world energy prices, world energy supply and demand, as well as US energy imports and exports. It is also correct that the market for these materials is not represented explicitly in NEMS.) In our model, the GHG emissions of these materials are part of the life-cycle assessment (LCA). Our single-region LCA model indeed assumes that vehicle production takes place in the US. We updated the text on lines 317–320 accordingly: “For simplicity purposes, the LCA model assumes that vehicle production takes place in the US [...] (see Supplementary Section 10 for a discussion of the error invoked from these assumptions).” Accordingly, we added a new section (Supplementary Section 10) and there we added: “Further, our single-region LCA model assumes that vehicle production takes place in the US. In reality,

many of the vehicles bought in the US are made elsewhere in the world. On the other hand, quite a few are exported.¹ We present evidence that the error invoked from disregarding these trade relations is small however. To that end we analyze the differences between the ‘full pricing’ and the ‘well-to-wheel’ pricing scenario. The ‘full pricing’ scenario fully prices embodied emissions of vehicle and battery production while the ‘well-to-wheel pricing’ scenario excludes pricing of vehicle and battery production emissions altogether. Hence, the cost of embodied vehicle and battery production emissions are zero under ‘well-to-wheel’ pricing. Yet, the differences in sales are marginal: merely a few longer-range EVs are partially replaced by shorter-range EVs (Figure S6h). Furthermore, these sales differences do not notably affect overall emissions outcomes (Figure 2a). As documented in Supplementary Table 2, the difference in total fleet-wide life cycle emissions between the two scenarios amounts to 71 Mt CO₂ cumulatively over the period 2010–2050. Hence, the results of this study are fairly unsusceptible to assumptions regarding the carbon intensity of vehicle and battery production which in turn partly depend on the location of production (inside versus outside of the US).”

In addition, the mentioned simplifications do not prevent us from capturing potential changes to material prices in our model, at least in a simplified way. Therefore, inspired by the reviewer’s thoughtful feedback, we added six new sensitivity cases, of which four assume constant prices of EV batteries from 2021 on. This constant price can be interpreted either as insufficient investments into battery technology or as growing raw material prices. IEA’s newly published World Energy Outlook² states that “a doubling of lithium or nickel prices would induce a 6% increase in battery costs. If both lithium and nickel prices were to double at the same time, this would offset all the anticipated unit cost reductions associated with a doubling of battery production capacity.” We note the results of this investigation in a new section called “Uncertainty analysis” as well as in Supplementary Section 6.

Likewise, in the LCA data, the ‘other’ materials would be expected to be important - otherwise the only(?) material difference between the vehicle types with functional relevance is copper? The remainder of the materials would appear to be structural? In this case, should the vehicle chassis be unified and a clearer delineation of the impacts that arise solely due to the

¹<https://legacy.trade.gov/td/otm/autostats.asp>

²<https://www.iea.org/reports/the-role-of-critical-minerals-in-clean-energy-transitions>

vehicle power train given? An inventory of the ‘other’ materials would be useful.

We apologize for not being fully clear in our previous draft. Accordingly, we added a caveat to lines 317–320: “For simplicity purposes [...] the model includes the most climate-relevant vehicle materials and disregards other minor materials (see Supplementary Section 10 for a discussion of the error invoked from these assumptions).”

In addition, we added to Supplementary Section 10: “our detailed process-based model allows for an explicit differentiation in composition and mass of the glider and power train. We consider the seven most common materials used for vehicle production: cast iron, stainless steel, automotive steel, wrought aluminum, cast aluminum, copper, and plastics. Combined these contribute more than 92% of the weight of the vehicle. The ‘other’ materials category is mostly comprised of glass and rubber, and for reasons of simplicity, we estimate the emissions of these other minor materials at 2 kg CO₂/kg material in the base year. This is well within the range of emission factors of rubber and glass which make up the vast majority of the ‘other’ category (by weight). For example, according to the ecoinvent 3.5 database using IPCC’s 2013 GWP-100 indicator, the production of synthetic rubber emits 2.75 kg CO₂/kg material, while natural rubber and uncoated flat glass emit 2.02 and 0.99 kg CO₂/kg.

Other potentially important materials specific to EV batteries may be cobalt, nickel and lithium, which are not considered in our model. Comparing our inventory and GHG emission results to that of a recently published and very detailed study,³ these materials account for 15% of the emissions associated with battery production. This translates into an omission of about 6% for the production of the vehicle as a whole, and of merely 1% for the entire EV life cycle including charging electricity in the model’s base year. We acknowledge that the relative error could be higher in future years assuming a decarbonization of the electricity mix but not the metal production. However, simultaneous improvements in battery technology are conceivable as well. We therefore anticipate that including the embodied CO₂ emissions from cobalt, nickel and lithium would not notably change scenario outcomes.” As mentioned above, this assertion is also confirmed if one analyzes the small differences in vehicle sales and resulting fleet life-cycle emissions between the ‘full pricing’ scenario and the ‘well-to-wheel pricing’ scenario (Figure 2a, Figure S6h, Supplementary Table 2).

³<https://link.springer.com/article/10.1007/s11027-019-09869-2>

Is there any consideration of the need for additional electricity generation to power a high BEV load in transport? (i.e. the electricity demand in the sector would increase, requiring additional capacity installation - not just a change from high CO₂ to low CO₂ mix)

We completely agree that this is important. Fortunately, the model indeed considers the additional electricity generation to power a high BEV load. In fact, this was one of the reasons why we chose the model. The effects of the additional electricity load should be clear from the additional information provided in the SI, such as Supplementary Figure 12, showing the increase in overall electricity demand, and Supplementary Figure 14, illustrating new additions to electricity generation capacity as well as retirements of old power plants.

It would be useful to include a diagram showing the flows of information and steps in the model to enhance understanding of the modelling approach.

We thank the reviewer for this suggestion. We now provide a graph showing an overview of the modelling framework in Supplementary Section 10.

Line 36 - is this 26% additional to the 1.5Gt emissions, or included?

The 26% are included in the 1.5 Gt. We double-checked the sentence to make sure that our language is clear.

Line 48 - aren't these models referred to as Integrated Assessment Models, not specifically Integrated Energy Models?

The definition of the term 'Integrated Assessment Model' is not consistent throughout the literature. Given that most of these models do not contain an assessment of the damage of climate change, we prefer to use the term 'Integrated Energy Model' for which some of the authors provided a detailed definition in a recent peer-reviewed paper.⁴ We also provide this definition in a new section in the SI (Supplementary Section 9).

Reviewer 2

This paper uses a quantitative model to simulate the transition from internal combustion engine vehicles to electric vehicles. The paper uses a life cycle analysis model of vehicle inputs and an energy model to look at how

⁴<https://www.sciencedirect.com/science/article/pii/S1361920919300513>

carbon pricing affects vehicle adoption. The paper shows that if renewables could provide more than 75% of electricity, then a carbon price on the extraction/processing of oil and the combustion of gasoline for conventional vehicles (and a similar carbon price for all other emissions like vehicle construction and the remaining polluting electricity sources) would lead to greater EV adoption than a carbon price just on gasoline combustion. The authors conclude that: “the simultaneous reduction of both direct and indirect emissions indicates a win-win situation for climate change mitigation, meaning that climate policy with very high shares of BEVs represents a no-regrets strategy.” And furthermore, that the US “should target deployment of BEVs and largely disregard competing technologies.”

Main comments:

1. Low prices of renewables are not sufficient for large scale adoption. Electricity reliability requires that supply equal demand at all moments. When the sun is not shining and the wind is not blowing, how does a grid with 75% renewables operate? Without explicit modelling of the very large costs of storage, additional transmission, and investments to address issues of reliability, the model likely understates the costs of, or even overstates the feasibility of, large-scale renewable deployment. A model that looks at the grid on an annual basis will not capture these issues.

We agree with you that reliability must be included in the analysis. Fortunately, NEMS includes a dispatch model that determines electricity supply, demand and prices at sub-annual level (three seasons by three times of day). It is important to note that NEMS explicitly models the cost of additional investments needed to allow for intermittent renewable electricity generation capacity. This is a core part of the capacity additions modeling. Thus, NEMS has the characteristics you identify as crucial for exploring our research question. We apologize if we were unclear and implied that NEMS was run only at an annual basis and did not model reliability. We have rectified this by adding the following sentence (lines 276–278): “A dispatch model determines electricity supply, demand and prices at sub-annual level (three seasons by three times of day).”

As for the feasibility of large-scale renewable development, we find that our results are backed by several previous studies indicating that such a high share of renewables

is possible. These studies even include work published in Nature Communications.⁵ ⁶
⁷ Of course achieving a high share of renewables is not the only way to achieve decarbonization of the electricity system. For example, see the careful discussion in the IPCC’s recent Special Report on Global Warming of 1.5 C, which is widely accepted by the scientific community. In the United States, the Biden administration aims to decarbonize electricity generation with a Clean Electricity Standard as the central policy, which will likely encourage development of renewables, but also opens the door to other technologies. These questions of the cost and feasibility of high renewables are very interesting, but they are not the research question at hand in our paper. Rather, we view the high renewables scenario as a starting point for our analysis.

2. Are fugitive methane emissions from extracting natural gas in the model?

We apologize for our missing description of methane emissions. We agree with you that methane emissions from fossil fuel extraction and transportation can be significant and should ideally be taken into consideration. Unfortunately, NEMS only reports emissions of CO₂ and a range of air pollutants. Thus, we have now made this very clear by updating the model description accordingly (lines 248–252): “Yale–NEMS provides a full account of CO₂ emissions across all industries and a range of air pollutants from vehicles and power plants. CO₂ accounted for 97% of total GHG emissions in the US electricity and transport sectors in 2019.⁸ Other GHGs such as methane emissions from fossil–fuel and hydroelectric power plants are not included.” We also now include a more detailed description of natural gas CO₂ emissions factors used in this work in our new Supplementary Section 6.

3. The authors’ conclusions ignore uncertainty. I suggest toning down the language like “no-regrets” and “disregard competing technologies”.

We agree with you; our previous language was too strong. In response to your comment, we deleted the clause “disregarding competing technologies”. In addition, we double–checked our language to make it more clear that BEVs are a “no regrets” strategy only if electricity decarbonizes, as has been assumed in our main scenarios. Examples include lines 20–22: “Given continued decarbonization of electricity supply, results show

⁵<https://www.nature.com/articles/s41467-019-13067-8>

⁶<https://www.nature.com/articles/s41467-020-18903-w>

⁷<https://www.nature.com/articles/s41467-020-16184-x>

⁸<https://www.epa.gov/ghgemissions/inventory-us-greenhouse-gas-emissions-and-sinks-1990-2019>

that a large-scale adoption of electric vehicles is able to reduce CO₂ emissions through more channels than previously expected”; lines 204–208: “In fact, the simultaneous reduction of both direct and indirect emissions indicates a win-win situation for climate change mitigation, meaning that climate policy with very high shares of BEVs represents a no-regrets strategy (but only if electricity continues to decarbonize as has been assumed in our main scenarios)”; and lines 404–407: “In terms of emissions, a carbon tax on supply chain emissions is not able to yield the desired results if the electricity grid does not face substantial decarbonization. In this case, pricing supply chain emissions leads to higher emissions compared to pricing direct emissions only (see Supplementary Section 6.1 for more details and results)”. Our new sensitivity cases which have been requested by the reviewer now make it clear that large adverse effects can occur if electricity slowly or barely decarbonizes.

4. In fact, showing how sensitive the paper’s results are to the assumptions on renewables penetration would be helpful. If renewables do not get cheaper and coal plants do not retire, what does the model find? I would think that the indirect emissions of extracting coal and natural gas for producing electricity (especially if fugitive emissions are high) could be similar to the range of the indirect emissions from extracting and processing crude oil.

We thank the reviewer for this thoughtful comment. As indicated above we include six new scenarios which explore the uncertainty in the costs of electric vehicles and renewable power plants (see section “Uncertainty analysis” and Supplementary Section 6). From the results of these scenarios it becomes apparent that the reviewer is correct in their assertion that the emissions from fossil-fuel based electricity could exceed those of gasoline production if the electricity sector fails to decarbonize substantially. In addition, we now provide a full list and brief description of our 28 scenarios in Supplementary Section 4. We believe that these 28 scenarios more than adequately explore the main uncertainties present in this work. The results of the extreme (i.e., most optimistic and most pessimistic) scenarios are described in detail throughout the main manuscript and the SI.

Minor comments:

5. Page 2, lines 34-35: gasoline production emissions presumably are in g CO₂/km, not kWh.

We chose to provide gasoline production emissions in g CO₂/kWh instead of g CO₂/km in order to avoid additional assumptions on vehicle fuel consumption (e.g., kWh/100km). We deeply considered providing g CO₂/km numbers along with the g CO₂/kWh numbers but we ask the reviewer for understanding that we refrained from this idea in the end. With the range of different vehicle segments and technologies analyzed in this work, we worry it would make the sentence too convoluted and thus difficult to communicate the key point to readers. However, we decided to provide values in g CO₂/gallon gasoline equivalents in Supplementary Section 1. We hope that this is an adequate solution and if this is a crucial point to you, we are open to reassessing.

6. Page 2, line 59 EIA stands for Energy Information Administration, not Agency.

Thank you for spotting this. We corrected accordingly.

7. Nature Communications does not use footnotes.

Thank you again. We made sure to avoid footnotes throughout the entire manuscript.

Reviewer 3

The submitted manuscript has investigated the life cycle environmental and economic implications of pricing, through the carbon tax, GHG emissions from passenger vehicles throughout their life cycles as an enviro-economical policy measure. In this pursuit, the researchers claim that pricing the GHG emissions associated with a vehicle's life cycle (though they have not provided a figure showing the system boundaries drawn for the life cycle assessment model) could be a more effective policy in reducing the environmental impacts of U.S. transportation than pricing only the GHG emissions from a vehicle's tailpipe. They further claim that such a policy measure could accelerate the phase-out of conventional vehicles while leading to higher penetration of battery electric vehicles and hence increased GHG savings. Environmental policies have not been as efficient and effective as they are supposed to be for mitigating the negative impacts of anthropogenic activities (e.g. transportation). Pricing vehicle's life cycle emissions through carbon tax is one of such policies that is likely to have far-reaching implications in terms of the sustainability profile

of U.S. passenger vehicles. Therefore, I think that gaining insights into these implications is of interest to the field of industrial ecology and others in the community and the wider field, especially given the significance of effective enviro-economical policies for mitigating the climate crisis studied in this field.

We deeply thank the reader for this assessment and very much share the view that this research is of interest to a wide range of research communities.

Two models - i) Yale-NEMS and ii) LCA model previously developed and published- were combined to carry out the quantitative analysis. Even though the supplementary information, including the input data, have been provided and includes data on many variables used in the models mentioned above, previously published works have been referred for some information, e.g. cost estimates for engines, electric motors, transmissions, fuel cells, and hydrogen storage tanks (and their specifications). The researchers are recommended to provide such information in a table, at least, in the SI.

We apologize for not having included this information previously. We now provide detailed cost figures for all modeled powertrain components in Supplementary Table 9 in the provided Excel spreadsheet. Similarly, detailed specifications of individual vehicles and vehicle components, including battery capacity and weight, battery depth of discharge, weight of motors, energy consumption, motorization, purchase price, fuel tank size, and total vehicle weight, have been included in Supplementary Table 11. Scenario-specific fleet-average fuel efficiencies, purchase prices, and degree of lightweighting are endogenous modelling results of NEMS, and have been documented in Supplementary Tables 5, 14 and 12.

Furthermore, even though the source code for the LCA model has been provided, the mathematical notations of the model formulations employed in the study, as well as of the incorporation of the assumptions into these models, are missing. I would recommend that the researchers consider addressing these in the SI, at least if the word limit does not allow them to be addressed in the main manuscript. This would make it easier for the reader to have a better understanding of the analytical work done and for other researchers to be able to reproduce the results and build upon the models. Also, the researchers are recommended to provide a figure, depicting the system boundary for the LCA model developed.

Thank you for this important suggestion. We now provide the three central equations of our approach in the methods section. For additional equations describing the LCA model and NEMS, interested readers can consult Wolfram et al.⁹ and the NEMS model documentation. We made sure that these references are shown more prominently in the methods section. In addition, as noted above, we now provide a figure of the modeling approach and the systems boundary in Supplementary Section 2.

I have submitted my comments as annotations and attached the annotated manuscript.

Line 30: The reference number 3 investigates the life cycle energy and environmental assessment of natural gas as transportation fuel in Pakistan and is not quite relevant for the statement it was given to support it. Given that the spatial and technological scope of the study is the United States, the reference number 3 can be replaced with a more relevant reference, e.g. having the same spatial scope such as Onat, N., Kucukvar, M., and Tatari O. (2014). Towards Life Cycle Sustainability Assessment of Alternative Passenger Vehicles. Sustainability 6 (12). Here, please, consider citing that work, instead, or adding that reference, as well, to acknowledge quite a relevant work.

We added this important reference.

Lines 31-32: This reads like an incomplete sentence and I could not understand. Please, consider revising this sentence.

We revised this sentence accordingly in order to make it more understandable: “These emissions occur off-site, or indirectly, and include generation of electricity to charge electric vehicles, in this work $\sim 66\text{--}86$ g CO₂ per electric-vehicle km driven in 2020, as well as the production of vehicles, here $\sim 16\text{--}38$ g CO₂ per vehicle-km driven in 2020 (Supplementary Section 1 and Supplementary Table 1).”

Line 34: Could the researchers explain why they preferred using such a unit when referring to the emissions from gasoline production? Did they use the conversion factor of 1 kWh = 0.03 GGE?

We use kWh because it is a unit listed in the International System of Units (SI). For better readability we would wish to stick to kWh in the main text. In Supplementary

⁹<https://www.sciencedirect.com/science/article/pii/S1361920919300513>

Section 1 however, we decided to also provide values in GGE using the conversion factor that the reviewer kindly provided.

Line 39: Since they are the main subject of the study, the researchers are recommended to indicate more clearly (maybe, in paranthesis) what these indicrect emissions include to get rid of the confusion caused by the ambiguity of the term.

Agreed. We changed the sentence accordingly. It now reads as follows (now lines 38–40): “The introduction of the Low Carbon Fuel Standard (LCFS) in California, which regulates all fuel and electricity production and combustion emissions, shows that transport policy in practice can at least partly address indirect vehicle emissions.”

Lines 44-45: The effect on production decision of changing costs due to regulatory standards such as CAFE has been investigated before. The researchers are recommended to refer to the study titled CAFE’s impact on the market share of electric vehicles by Sen et al. (2017) to ackonwledge a relevant work.

We included this interesting reference.

Line 75: What is meant by this term? Please, elaborate.

We changed the sentence to make it more clear. It now reads (now lines 77–79): “Here we address these knowledge gaps by applying a novel conceptual framework by Creutzig et al., which focuses on energy–demand side (rather than energy–supply side) solutions to climate change mitigation.”

Lines 79-80: These sectors are already parts of an EV’s life cycle under the IO modeling setting (which can be represented as unit processes under the process modeling settin). Such responses are reflected upon in the life cycle sustainability assessment studies of different vehicle classes, if I have understood this sentence, correclly. That’s why the researchers are recommended to clarify what are meant by direct and indirect emissions and by ‘vehicle sectors’, and provide a figure, depicting the system boundary adopted for this study.

The reviewer is absolutely correct in stating that the sectors material production, vehicle manufacturing and electric charging are reflected and adequately linked to each other in the LCA model. On the other hand, NEMS has the advantage of better captur-

ing the non-linear dynamics of the entire energy system including changes in electricity supply. In order to make use of the advantages of both models, we soft-link our LCA model to NEMS. We hope that this becomes clear from our description, especially now that we included a new figure depicting the system boundary of this work (Supplementary Section 2).

Line 87: So, Scenario 1 assumes that the emissions from the tailpipe are accounted for, priced, and optimized for, whereas Scenario 2 assumes that the emissions from the entire vehicle supply chain are accounted for, priced, and optimized for. Is that correct?

Yes, the reviewer is correct. We double-checked our scenario descriptions throughout the main manuscript and the SI and we hope that they are adequate and clear.

Line 95: Does “direct tailpipe emissions” mean that tailpipe emissions are the direct emissions? I believe there is a need to provide a clarification as to what are meant by direct and indirect emissions. Are direct emissions those from tailpipe or from vehicle sectors? Similarly, are indirect emissions those from non-tailpipe emissions or from material extraction sector? The researchers are recommended to clearly define these emissions to avoid any confusion.

The reviewer interpreted correctly what is meant by ‘direct tailpipe emissions’. In order to avoid any ambiguities we state on line 35 that we use the term ‘direct emissions’ and ‘tailpipe emissions’ as synonyms.

Line 104: Is this referring to the “well to tank” emissions? Please elaborate.

Energy-chain emissions describe emissions invoked by the production and use of energy carriers (well-to-wheel). We added to line 104 (now line 107) that the term ‘energy-chain’ emissions is synonymous with ‘well-to-wheel’ emissions in order to avoid confusion.

Lines 104-105: The researchers assumed that hydrogen production becomes carbon-neutral by 2050 through the hydrogen production from biomethane, with CCS. Why have the researchers not considered the green hydrogen production, at all? How do they think that such a consideration would affect the conclusions of the study?

In Supplementary Section 1 we provide an example on how carbon neutral produc-

tion of hydrogen could be achieved. We added the reviewer’s example of a green hydrogen pathway to the text. We also explain how this pathway would influence modelling outcomes. The section now reads as follows: “Net-zero could be achieved in various different ways, for example by producing hydrogen exclusively from wind or solar power or by using a hydrogen production mix consisting primarily of hydrogen from biomethane with carbon capture and storage, representing a carbon sink with about -36 g CO₂e/kWh, and a small remainder, around 7–8%, of hydrogen from SMR, emitting around 450 g. Since carbon-neutral hydrogen production is only considered in one of our side cases it has been modeled in less detail: For the carbon-neutral hydrogen side case NEMS only receives the hydrogen production emissions factor from the LCA model, which can be assumed equal under both production pathways. Hence, for the purpose of this paper, whether hydrogen is produced from renewables or from a combination of biomethane CCS and natural gas, has no effect on the emissions outcomes because both production pathways would ensure net-zero emissions.”

Lines 113-114: Here the researchers are recommended to cite a study on the material footprint of electric vehicles through MRIO, conducted by Sen et al. (2019). This is one of the very few studies in the literature that shows the material intensity of EVs, which is very relevant to cite to acknowledge the previous work in this regard.

Thank you for this suggestion. We added the reference.

Line 116: Are these ones different from the a, b, and c previously mentioned?

They are in fact the same. In order to avoid confusion we added a reference to Supplementary Figure 6 and changed the sentence. It now reads (now lines 120–123): “As mentioned earlier, we explore a range of side cases (Supplementary Section 3) which show some variation in their potential for emission reductions (also see dotted lines in Figure 2a–j) but the overall trend is robust among these cases.”

Line 152: The researchers are recommended to provide provide the percentages, as well, to help the reader relate better.

We made sure that percentages are provided as well. Please see the previous paragraph on lines 151 to 158. In addition, relative changes are also indicated in Figure 4. We would also like to note that all underlying data for Figure 4 is included in Supplementary Table 7 in the accompanying Excel spreadsheet.

Line 172: Given how significantly battery manufacturing influences the sustainability impacts of EVs, why have the researchers not considered discussing the implications of such a policy measure in terms of the future of EV batteries and how increasing demand on BEVs would likely influence battery technologies and the demand for them, as well as their prices?

We agree with the reviewer that EV batteries have a significant influence on sustainability issues. Our focal point is however not on emissions from batteries in particular but rather on the overall influence of indirect emissions on optimal vehicle fleets. We ask for the reviewer's understanding that, due to the limited space in the main manuscript, we had to focus on a high level discussion of the topic. However, in response to your comment, we added some additional discussion items on vehicle and battery materials to Supplementary Section 10.

Lines 187–188: This is a common finding of the studies on vehicle LCA that do not consider pricing, at all. So, how does the pricing affect this? Through a higher market penetration of BEVs thanks to taxing the full LC emissions that influence the consumer's purchase behavior?

The reviewer is correct again. A useful aspect of our study is the consideration of prices and the ability to take consumer decisions into account. As the reviewer points out correctly, this ability of the model enables us to study the effects of holistic pricing of all embodied emissions of different vehicle options. The model finds that a pricing of all emission sources leads to a different optimal solution than pricing of tailpipe emissions alone.

Line 189: Does this refer to the extraction of crude oil?

We thank the reviewer for spotting this. In fact, we wanted to refer to the entire production process of crude oil, including both exploration and extraction. In order to avoid any ambiguities we rephrased the sentence. It now reads as follows (now lines 197–198): “However, higher electricity emissions are more than offset by lower gasoline supply-chain emissions stemming from the production of crude oil (Figure 2k).”

Line 256: The researchers are recommended to provide these assumptions/values, along with their references, in a table. In fact, given the large scope of the study, it might even be better to provide all your assumptions in a table.

We agree with you completely and fully support the reproducibility of scientific re-

sults. We therefore provide all cost assumptions of various vehicle components in Supplementary Table 9 in the provided Excel spreadsheet. In the same spreadsheet we provide 21 additional data tabs representing an exhaustive modelling database.

Line 269: The researchers are recommended to consider normalizing Fig. 8b or present a separate figure, with normalized price information per BTU, for example.

We provide an additional panel in Supplementary Figure 8 depicting normalized gasoline and electricity prices per BTU as requested by the reviewer. For conversion we use conversion factors provided by the U.S. EIA.¹⁰

Lines 273–275: How was this formulated and incorporated into the model? The researchers are recommended to consider providing the mathematical notations of all their formulations.

In this instance, apart from updating the costs of rooftop solar PVs, no changes have been made to the NEMS model by the authors. Due to space constraints unfortunately we cannot reproduce the detailed equations and descriptions contained in the NEMS modelling documentation. However, we added a reference to the specific section of the NEMS documentation (section “Distributed Generation and Combined Heat and Power (CHP) Submodule” within the “Commercial Demand” section of the NEMS documentation). Interested readers will be able to find all the required information there. Please also refer to the new figure in Supplementary Section 2 where we depict a simplified representation of the commercial and residential buildings sectors.

Lines 276–279: Are these the researchers’ assumptions?

This paragraph describes future electricity demand growth, the development of electricity emissions as well as the resulting electricity carbon intensity, all of which are NEMS modelling results. These developments are a direct result of our cost assumptions on renewable electricity generators. We double-checked lines 274 to 277 to make sure that these assumptions are clearly explained there.

Line 303: What do C\$, LCC\$, WTW\$... mean, and what do the values provided represent, exactly?

¹⁰<https://www.eia.gov/energyexplained/units-and-calculators/british-thermal-units.php>

We apologize for this mishap. These acronyms used in the supplementary Excel spreadsheet are only meant for internal use. We renamed them accordingly in order to be consistent with the scenario names in the manuscript. We made sure that the first tab of the Excel spreadsheet file describes the content of each following tab. In addition, we double-checked that each tab has a descriptive header and that all units are provided.

Lines 309–310: Overall, I think there is a lack of mathematical representation of the model(s) formulations and of the incorporation of assumptions into these formulas. Please consider providing the mathematical notations.

As mentioned above, we now provide the three central equations of our approach in the methods section. For additional equations describing the LCA model as well as NEMS, interested readers can consult Wolfram et al.¹¹ as well as the NEMS model documentation. We made sure that these references are shown more prominently in the methods section.

Line 327: Why have the battery capacities been assumed constant after 2025? And, have the researchers assumed any improvements in the GHG efficiency of battery production over the years?

In order to address these important questions, we added two new sections (Supplementary Sections 6.2 and 11). We address the first part of the question in Supplementary Section 6.2: “A standard assumption of Yale–NEMS is that of constant EV battery capacities (as well as constant battery weights and densities and hence EV ranges) after 2025 (Supplementary Table 11). In most of our scenario runs we adopted this assumption because, on the one hand, batteries could become more energetically dense in the future, hence requiring smaller capacities. On the other hand, larger capacities may be needed if BEVs continue to increase in driving range. Both factors could cancel each other out, hence the constant capacity assumption. However, in this section we present two sensitivity cases that explore the effects of increasing battery densities, leading to smaller, lighter, less material-intensive and cheaper EV batteries while providing the same driving range. We assume that – averaged over all technologies – battery densities continue to increase by about 1.5% per year after 2025. This rate is somewhat higher than the assumed average 2010–2025 increase of about 0.9% per year. This development helps especially with cost reductions of longer-range BEVs. This is true under both direct-emissions-only pricing

¹¹<https://www.sciencedirect.com/science/article/pii/S1361920919300513>

and marginally more so under full-emissions pricing because the emissions penalty of the smaller batteries is also lowered. Hence, a shift in sales from 100-mile to 200-mile range BEVs can be observed. In addition, this trend is accompanied by an overall increase in BEV sales shares (Figure S10). ”

The second part of the question is addressed in Supplementary Section 11: “The GHG intensity of batteries in particular (and vehicle production in general) improves in several ways in our model. The main scenarios assume a falling carbon intensity of electricity, which reduces both the emissions during the material production stage as well as emissions invoked during the battery assembly stage (Supplementary Table 18). The assumed energy mix of the material production and battery assembly stages is comprised of heat from fossil fuels as well as electricity (see Supplementary Tables 19, 20). In addition, some scenarios assume improved recycling of materials and reuse of components such as batteries (see Supplementary Tables 21, 22), further reducing GHG emissions.” We also hope that the figure provided in Supplementary Section 2 further helps to communicate these relationships.

Reviewer comments, second round review -

Reviewer #1 (Remarks to the Author):

The paper has been much improved.

I would prefer that the figure describing the modelling was in the main text rather than the supplementary material, as it would better give context to the results.

The paper as a whole tends to lean on the supplementary material to a significant extent, which detracts from understanding - perhaps the authors could add some additional commentary to the supplementary material so that it could be read more as a report, so that those who are interested in understanding in detail could skip the paper and go to the supplement.

Reviewer #2 (Remarks to the Author):

Thank you for addressing my previous concerns. I have no further comments.

Reviewer #3 (Remarks to the Author):

I thank the authors for addressing and clarifying all my comments and concerns regarding the submitted manuscript. Nature, as a leading multidisciplinary science journal, is a source of scientific knowledge that is highly trusted both by academia and society, in general. Every single sentence that is written in any manuscript that is to be potentially published in Nature must be paid due attention, and all the assumptions considered in any analysis must be justified e.g. by means of providing relevant publication(s), accordingly. So, as my last comment, I suggest that the authors make an extra effort to highlight, at least, any significant assumptions throughout the manuscript that will enhance the understanding of the readers, especially the primary target audience.

Overall, given the comment above, the manuscript is suggested for publication after that very minor revision.

October 27, 2021

Response to the reviewers

Dear anonymous reviewers,

We would like to thank you again for reviewing our revised manuscript entitled *“Pricing indirect emissions accelerates low-carbon transition of US light vehicle sector”*. We incorporated all of your remaining suggestions and look forward to hearing from you.

Kindest regards,

Paul Wolfram (on behalf of all authors)

Reviewer 1

The paper has been much improved. I would prefer that the figure describing the modelling was in the main text rather than the supplementary material, as it would better give context to the results. The paper as a whole tends to lean on the supplementary material to a significant extent, which detracts from understanding - perhaps the authors could add some additional commentary to the supplementary material so that it could be read more as a report, so that those who are interested in understanding in detail could skip the paper and go to the supplement.

Thank you for this comment. In response to your suggestion, we copied a simplified, more compact version of the figure describing the modelling framework into the main manuscript. The full version of the figure would require many more words in the main text, muddling the flow of the manuscript and requiring cuts elsewhere, but we believe that the compact version allows us to convey the main ideas in a concise way. We also added a new section to the beginning of the supplementary material briefly summarizing the results of the paper. In addition, we strictly adhered to the style guidelines of Nature Communications when preparing the supplementary material.

Reviewer 2

Thank you for addressing my previous concerns. I have no further comments.

Thank you again!

Reviewer 3

I thank the authors for addressing and clarifying all my comments and concerns regarding the submitted manuscript. Nature, as a leading multidisciplinary science journal, is a source of scientific knowledge that is highly trusted both by academia and society, in general. Every single sentence that is written in any manuscript that is to be potentially published in Nature must be paid due attention, and all the assumptions considered in any analysis must be justified e.g. by means of providing relevant publication(s), accordingly. So, as my last comment, I suggest that the authors make an extra effort to highlight, at least, any significant assumptions throughout the manuscript that will enhance the understanding of the readers, especially the primary target audience. Overall, given the comment above, the manuscript is suggested for publication after that very minor revision.

Thank you for this suggestion. We made several final changes that further improved the readability and the understanding of our manuscript. First, we restructured the introduction. Second, we included a simplified version of Figure S3 in the main text. Finally, we carefully double-checked each and every sentence and made sure that all assumptions made in our work are clearly documented. We strongly believe that the manuscript is now ready for publication in Nature Communications.